# The Effect of Lameness on Milk Production of Dairy Goats

**DOI:** 10.3390/ani13111728

**Published:** 2023-05-23

**Authors:** Natasha Jaques, Sally-Anne Turner, Emilie Vallée, Cord Heuer, Nicolas Lopez-Villalobos

**Affiliations:** 1School of Agriculture and Environment, Massey University, Palmerston North 4442, New Zealand; n.lopez-villalobos@massey.ac.nz; 2Dairy Goat Co-Operative (NZ) Ltd., 18 Gallagher Drive, Melville, Hamilton 3206, New Zealand; sallyanne.turner@fonterra.com; 3EpiCentre, School of Veterinary Science, Private Bag 11-222, Palmerston North 4442, New Zealand; e.vallee@massey.ac.nz (E.V.); c.heuer@massey.ac.nz (C.H.)

**Keywords:** goat, dairy, milk production, fat, protein, lactose, ratio, lameness, welfare, income

## Abstract

**Simple Summary:**

Lameness on dairy goat farms is a welfare concern and could negatively affect milk production. The aim of this study was to quantify the effects of clinical lameness on dairy goat milk production from three commercial goat farms in New Zealand. Goats that were severely lame (walking on three legs) produced between 7.05 and 8.67% less milk than goats that were not lame. When the prevalence of severe lameness was between 5 and 20% of the herd, the estimated average daily milk income lost was between NZD 19.5 and 104 per day. This study established the negative impact of lameness on milk production and the loss of annual income from lame dairy goats on three commercial farms.

**Abstract:**

Lameness on dairy goat farms is a welfare concern and could negatively affect milk production. This study’s objective was to evaluate the effects of clinical lameness on the daily milk production of dairy goats. Between July 2019 and June 2020, 11,847 test-day records were collected from 3145 goats on three farms in New Zealand. Locomotion scoring of goats used a five-point scoring system (0 to 4). The dataset was split into two groups by lactation type, where goats were classified as being in seasonal lactation (≤305 days in milk) or extended lactation (>305 days in milk). A linear mixed model was used to analyze datasets using milk characteristics as the dependent variables. Severely lame goats (score 4) in seasonal and extended lactation produced 7.05% and 8.67% less milk than goats not lame, respectively. When the prevalence of severe lameness is between 5 and 20% of the herd, the estimated average daily milk income lost was between NZD 19.5 and 104 per day. This study established the negative impact of lameness on milk production and annual income in dairy goats on three farms.

## 1. Introduction

Animal welfare issues on dairy goat farms, such as lameness, have increasingly been highlighted in past studies conducted on goat farms worldwide [1,2,3]. The within-herd prevalence of lameness on some European commercial dairy goat farms ranged from 1.7 to 67% over the last 20 years [1,4,5]. Despite the large variation in prevalence between studies, lameness on commercial farms is a global problem. While lameness prevalence has been quantified, the effects of lameness on dairy goats have been largely overlooked in the past. Increased productivity and animal efficiency would improve resource utilization, which is necessary because of future competition over scarce resources [6].

Lameness has economic consequences for farmers and has been extensively researched in dairy cattle [7,8,9] and sheep [10,11,12], but not in dairy goats. In dairy cattle, lameness is the second most costly animal health problem after mastitis [13]. In the New Zealand dairy cattle industry, the estimated lameness cost for farmers was NZD 94.00 per cow [14]. This study was published over 30 years ago, so the cost will likely be higher now. More recently, a U.S. study estimated that digital dermatitis and white line disease, both causes of lameness, cost farmers NZD 94.00 and NZD 234.89, respectively [15,16].

In the sheep industry, footrot, an infectious cause of lameness, is the second most costly health problem after gastrointestinal parasites. In the New Zealand and British sheep industries, lameness caused by footrot has cost the industry an estimated NZD 9 million per year and GPB 24 million (NZD 48 million) per year, respectively [10,11]. In a more recent British study of sheep farms, it was reported that lameness management costs the farmers between GPB 3.90 and 6.35 (NZD 7.45 and 12.1) per ewe per year, depending on the lameness prevalence within the herd [17]. These costs estimate lameness’ impact on the dairy cow and sheep industries. Extrapolation of this research to dairy goat studies should be used cautiously because dairy goats on commercial farms are commonly housed indoors and managed differently from dairy cows and sheep [18]. Therefore, it is currently unknown how lameness may impact dairy goat farmers economically.

Literature on the effects of lameness in dairy goats and sheep is scarce. Though not yet quantified, lameness in dairy goats has been linked to reduced milk yield, fertility, and longevity [19,20,21]. Lameness in dairy sheep was also significantly associated with a reduction in milk yield [22]. Additionally, Gelasakis et al. [22] reported that the effect of lameness in high-yielding sheep was associated with greater milk losses than in control sheep. In contrast to dairy cows and dairy sheep, in the present study, it is hypothesized that the milk yield of dairy goats was negatively affected by lameness and claw disorders [23,24]. No goat studies have quantified the association between milk production and lameness. This study aims to evaluate the effect of lameness on milk production and milk characteristics and the potential loss of income on three commercial farms based in New Zealand.

## 2. Materials and Methods

The animal study protocol was approved by the Ethics Committee of Massey University, New Zealand (MUAEC Protocol 19/51, 29 May 2019).

### 2.1. Data Collection

Data collection consisted of records from 3145 goats from three farms based in Waikato, New Zealand. The goats were a combination of Saanen, Toggenburg, and Alpine breeds, and various crosses, which are generic to the New Zealand goat industry [25]. As the goat breed was not accurately recorded, the goat breed was not included in the analyses. Goats were housed in semi-indoor conditions and kidded once a year. Seasonal lactation goats kidded between June and August 2019 (≤305 days in milk). All three farms also had a group of goats undergoing extended lactation (>305 days in milk). Extended lactation goats were not bred at seasonal breeding periods or had not conceived but were retained in the herd and were continuously milked for prolonged periods.

Within each farm, seasonal and extended lactation goats were housed and managed in the same manner. Two of the three farms mixed the seasonal and extended lactation goats together, while the third farm kept the groups in separate pens within the same barn. Goats were milked twice a day and fed ad libitum with a total mixed ration and fresh grass cut and carried from nearby paddocks. The total mixed ration was a composition of minerals, grass silage, and sometimes maize silage. The total mixed ration and fresh grass were fed all year round, though the ratio to each other may have varied depending on the time of year owing to the variable grass growth throughout the different seasons. A supplementary concentrate meal, distillers dried grain (DDG), was given in the milking parlor.

Locomotion scoring events were carried out five times for farm A and four times for farms B and C across one lactation from July 2019 to June 2020. Lameness was scored using a five-point locomotion scale developed by Deeming et al. [26] with minor modifications (Appendix A). Briefly, the scores were defined as 0—normal, 1—uneven, 2—mildly lame, 3—moderately lame, and 4—severely lame. A goat was classified as clinically lame if its locomotion was scored as a three or a four. In relation to herd-testing events (milk collection events), lameness scoring events occurred either at the same time or within a few days of the herd-testing events. There was only one occasion where the herd test was a few weeks after the locomotion scoring event. This was corrected in the analysis by adding the variable, the date difference between the locomotion scoring event and the herd test event, into the statistical analysis.

### 2.2. Statistical Analysis

All analyses were conducted in SAS (version 9.4, SAS Institute Inc., Cary, NC, USA) using the MIXED procedure.

The primary dataset was split by type of lactation, seasonal and extended lactation goats. The seasonal lactation dataset comprised 1782 goats with 6368 test-day records. The extended lactation dataset comprised 1363 goats with 5479 test-day records. Univariate mixed linear models were conducted to determine which variables were significantly associated with test-day milk production characteristics. The dependent variables included daily yields of milk, fat, protein, and lactose; concentrations of fat, protein, and lactose; fat/protein ratio; somatic cell score; and milk income. The somatic cell score was calculated as average log2 (somatic cell count/1000). The evaluated milk income assumed a payment for the producers of NZD 19 per kg of milk solids, where milk solids were the sum of fat, protein, and lactose yields.

The linear model for the analysis of seasonal lactation goats included the fixed effects of lameness score, parity, stage of lactation, deviation from the median kidding date, estimated breeding values for lactation yields of milk, fat and protein, date difference between the locomotion scoring event and the herd test event, and the random effects of test-day and residual error. There was a significant interaction between parity and stage of lactation for all of the dependent variables except for the fat/protein ratio, which had a significant interaction between lameness score and stage of lactation.

The fixed and random effects used for the extended lactation goats were the same as the seasonal lactation goats, except for three differences in the variables used. Firstly, within the extended lactation group of goats, there was considerable variation in lactation lengths (ranging from 272 to 3738 (>10 years) days in milk at the first test-day event). Because of this, season of the year was used as a factor rather than the stage of lactation. Season of the year had two levels, winter-spring (June 2019–November 2019) and summer-autumn (December 2019–May 2020). Secondly, the deviation from the average kidding date was omitted from the model because the deviation from the median kidding date in this group ranged from 1 to 3628 days (≈10 years); therefore, its effect was considered to be low and not important in this analysis. Lastly, the only significant interaction included in the models was between the lameness score and parity when the daily yields and the milk income were dependent variables.

The random effects of the farm herd test date, animal, and residual were assumed with zero means and variances, $\sigma_{f}^{2}$, $\sigma_{a}^{2}$, and $\sigma_{e}^{2}$, respectively. A repeated effect with a serial autocorrelation was not included in the analysis as it was assumed to be low because there were 3 to 4 months between herd-testing events.

The marginal means for each fixed effect level were used for multiple comparisons with adjustment by the Tukey–Kramer method [27].

A hypothetical scenario of one herd of 1000 goats was used to illustrate the average daily milk income lost at the different prevalence levels of severe lameness. The following equations were used:Income lost = (percentage of goats severely lame) × (difference in milk income) × 1000,(1)
where milk income assumed a payment for the producers of NZD 19 per kg of milk solids, where milk solids were the sum of fat, protein, and lactose yields, and
Difference in milk income = (average income from goats with a locomotion score of 0) − (average income from goats with a locomotion score of 4) × 1000 (2)

## 3. Results

The percentage of the goats across parity, locomotion score, and clinical lameness by type of lactation, seasonal and extended lactation, are shown in Table 1. Most seasonal lactation goats (66%) were of parity 1 or 2. Most extended lactation goats had given birth only once before entering the continuous lactation group. Across the 2019–2020 production year, 24.4% and 46.3% of seasonal and extended lactation goats were clinically lame at least once, respectively.

Seasonal and extended lactation goats differed in average milk production characteristics (Table 2). Average milk characteristics were higher for seasonal lactation goats than extended lactation goats, except for fat and protein percentages and somatic cell counts.

### Effect of Lameness on Daily Milk Yield and Composition

For seasonal lactation goats, when assessing the effects of locomotion scores on milk production and characteristics, the greatest reduction was between goats with a normal gait (score 0, Table 3) and goats defined as severely lame (score 4). The reduction in milk, protein, and lactose yield was significant (*p* < 0.05). Milk yield reduced by 7.05%, protein yield decreased by 8.26%, lactose yield decreased by 7.06%, and milk income reduced by 6.69%.

For extended lactation goats, all milk characteristics were significantly (*p* < 0.05) reduced for goats with severe locomotion impairment compared with goats that were not lame (Table 4). There was a reduction of 8.67% for milk, 4.81% for fat, 8.48% for protein, and 7.69% for lactose yields. Consequently, the average milk income was 6.18% lower.

When severe lameness was present within a herd, the reduction in milk income depended on whether the goats were in seasonal or extended lactation (Figure 1).

In the scenario where the prevalence of lameness increased, the daily loss of income was higher when the prevalence of severe lameness was high, regardless of lactation type. The cost of severe lameness was higher in seasonal goats than in extended lactation goats. For example, a seasonal herd with 200 severely lame goats would cost the farmer NZD 104 per day, while an extended lactation herd with the same number of severely lame goats would cost the farmer less, with a loss of NZD 78 per day.

## 4. Discussion

The average milk production yields for dairy goats within the study farms were within the range previously reported in New Zealand [21,28,29]. Milk component concentrations also fell within the range of national [29] and international studies [30,31]. Variations were most likely due to management and environmental factors [29,32].

### Association of Lameness with Milk Production

This study has highlighted that severely lame goats (score 4), regardless of lactation type, had significantly lower milk production than goats that were not lame. The impact of lameness on milk production differed from dairy sheep and cow studies. Assuming that locomotion scoring was on the day of diagnosis, the estimated daily reduction in milk yield in severely lame seasonal and extended lactation dairy goats was 7.05 and 8.67%, respectively. The total annual loss for a herd would depend on the average prevalence of clinical lameness and the average duration of the clinical lameness. In dairy sheep, the daily milk yield was reduced by 10.8 to 35.8%, depending on when lameness was diagnosed [22]. Gelasakis et al. [22] reported that lameness affected milk production though the effect depending on the time of diagnosis during a lameness episode. A 10.8, 32.5, and 35.8% reduction was recorded two weeks before, during the week, and one week after the lameness diagnosis in those sheep, respectively. The present study did not relate the relative loss to the time of diagnosis. Instead, the estimated milk reduction applied to an ‘average day’ of lameness throughout the season. Therefore, the actual loss of production is most likely higher than what has been estimated in this study.

In dairy cattle, reduced milk yields in lame cows were reported [33,34]. Similar to this study, dairy cows’ daily loss of milk yield was between 0.82 and 11.1% [33,35,36]. Over an entire lactation (305 days), milk yield reduction was between 2.38 and 9.92%, depending on the cause of lameness [34]. In dairy cows, reduced milk yields were apparent for up to four months before and five months after diagnosis [37,38]. Another study on dairy cows reported that lame cows had a reduced test-day milk yield until up to eight months after diagnosis [9]. In addition, there are reports that lame cows had higher production than healthy cows before the onset of lameness [34,38]. This would imply that high production predisposed cows to lameness. Owing to only one milk test-day event occurring around the time of the locomotion scoring in the current study, the impact of milk lost for a goat over one lameness event should be investigated further in dairy goats to study the relationship between lameness duration and milk production. Additionally, it would be important to determine if high genetic merit goats had an increased risk of becoming lame.

The effect of lameness on milk production would be the function of the duration and severity of a case of lameness. The longer the goat was left untreated, the more severe and painful the conditions became, along with an increase in production losses due to the longer recovery. As Gelasakis et al. [22] suggested in their study on lame sheep, lame goats were probably less able to compete for high quality and quantity of feed, despite the goats having had ample headspace at the feeding passage (330 mm per head). Like cows, a goat in pain may have drifted lower in the herd’s hierarchy and be outcompeted by non-lame goats when the ration is fed out [39]. Alternatively, inflammatory factors could negatively impact the goat’s appetite and reduce milk production [22].

The reduction in milk production of severely lame goats was comparable to milk production loss in mildly to severely lame cows [38]. A possible reason for a relatively lower loss in goats with less severe cases of lameness was that they were housed indoors on soft bedding all year and could lie down next to the feeding passage to feed. Thus, lame goats might find it easier to meet their daily nutritional needs, unlike dairy cows or sheep that usually graze outside for all or part of the year and would have to walk more than goats to meet their daily nutritional requirements. Furthermore, cows housed indoors still have to walk to and stand at the feeding passage, while dairy goats on commercial farms do not have to walk or stand to feed themselves. A comparison of the time spent lying down and their geographic location in the barn between lame and non-lame goats should be investigated further.

The present study is the first to distinguish between seasonal and extended lactation goats’ production and their response to lameness. The reduction in milk production was higher in the extended lactation goats than in seasonal lactation goats; however, the reduction in income of clinical lameness to the farmer was higher in seasonal lactation goats. Clinically lame and severely lame goats in seasonal lactations had a reduction of 2.63 and 7.10%, respectively, in average milk production compared with goats that were not lame. The milk production of clinically lame and severely lame goats in extended lactation was reduced on average by 4.66 and 8.56%, respectively. The physiological differences between seasonal and extended lactation goats are unknown; therefore, some underlying factors could differentiate the goats’ response to lameness and the subsequent effect on their production. Another reason extended lactation goats had a higher milk yield reduction could be farm management factors. For example, if seasonal goats breed each year, the farmer could prioritize the treatment of seasonal goats above extended lactation goats. Therefore, delaying treatment could increase the severity of lameness in extended lactation goats. Further investigation needs to determine whether this difference is physiological or because of management.

Lameness on U.K. commercial dairy goat farms appears to be a low priority for most farmers [40]. The farmer survey of that study reported that kidding health, Johnne’s disease, tuberculosis, and nutrition were more important production-limiting factors than lameness. Lameness appeared less important to farmers because they generally underestimated their herd’s lameness prevalence and tended to accept mild lameness within their farm as an unavoidable phenomenon [1,41]. The costs of lameness events are not always directly observed; therefore, farmers may often be unaware of the full extent of the economic impact associated with lameness [41].

When the farm gate price for milk is relatively high, farmers may tend to accept the costs due to lameness. In our estimates, milk income was substantially reduced when goats were clinically lame and further when the goats were severely lame. Farmers within the study incurred regular management costs for preventing or treating lameness, including hoof trimming, feed supplements, footbath supplies, and direct costs like labor and treatment. Unlike dairy cow farmers, it is uncommon for commercial dairy goat farmers to call a veterinarian to treat individual goats for lameness. These factors must be accounted for when undertaking an economic analysis of lameness on farms, which would be the next step forward after this study. Therefore, depending on the incidence rate of lameness per year, the cause and type of lameness, the number of treatments required, the number of herd hoof trimmings, and other prevention strategies, the cost of lameness for farmers will vary substantially. The resulting financial burden can severely impact the farmers’ production costs and subsequent profits.

## 5. Conclusions

This is the first study that quantified the negative effect of lameness on milk production in dairy goats on commercial farms. Milk production was significantly affected, where severe lameness resulted in a 7.05% decline in milk production for seasonally lactating goats and 8.67% for goats in extended lactation. Fat, protein, and lactose concentrations were other milk characteristics significantly affected by severe lameness, where fat and protein concentrations in severely lame goats became elevated with a significant reduction in milk yield. At a relatively high prevalence of severe lameness, a reduction in milk income can have grave impacts on the economic efficiency of a commercial dairy goat farm.

## Figures and Tables

**Figure 1 animals-13-01728-f001:**
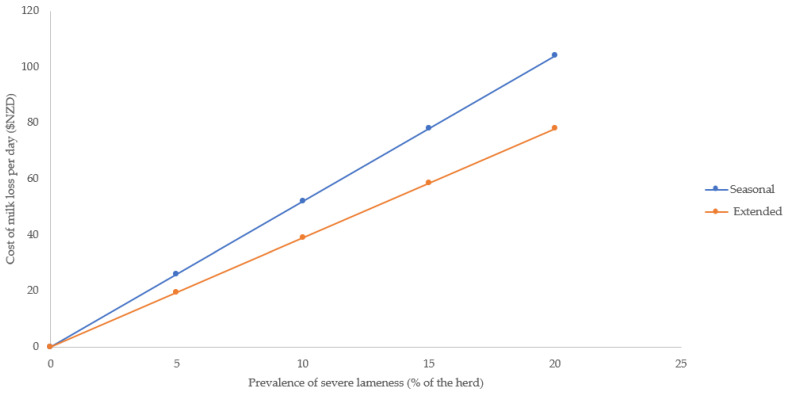
The average daily loss of income due to a reduction in milk solids (sum of fat, protein, and lactose yields) at different prevalence levels of severe lameness (locomotion score 4) on a farm with 1000 goats and an income from the sale of milk solids at a farm gate price of NZD 19.00/kg for seasonal and extended lactation goats. Note that these averages did not take into account the duration of lameness.

**Table 1 animals-13-01728-t001:** Percentages of goats by parity and clinical lameness status (locomotion score 3 or 4) for seasonal (≤305 days in milk) and extended (>305 days in milk) lactation groups from three farms in New Zealand during the production season of 2019–2020.

Variable	Seasonal Goats (*n* = 1782)	Extended Goats (*n* = 1363)
Parity 1	41.4	60.1
Parity 2	24.3	18.1
Parity 3	17.8	12.5
Parity 4+ ^1^	16.5	9.40
Goats clinically lame	24.4	46.3

^1^ Goats 4 years old and older.

**Table 2 animals-13-01728-t002:** Descriptive statistics of daily milk characteristics by seasonal (≤305 days in milk) and extended (>305 days in milk) lactation groups on three farms in New Zealand during the 2019–2020 production season.

Test-day Milk Characteristics	Seasonally Lactating Goats (*n* = 1782)		Extended Lactation Goats (*n* = 1363)	
	Mean	Standard Deviation	Mean	Standard Deviation
Daily yields				
Milk (kg)	3.75	1.13	3.14	0.99
Fat (g)	117	39.0	103	32.9
Protein (g)	118	35.3	103	30.9
Lactose (g)	170	53.4	140	45.1
Concentration (%)				
Protein	3.17	0.34	3.32	0.34
Fat	3.14	0.55	3.31	0.58
Lactose	4.54	0.27	4.47	0.29
Fat/protein	0.995	0.17	1.000	0.16
Somatic cell score ^1^	9.50	1.42	10.0	1.15
Milk income ($/kg) ^2^	7.70	2.34	6.57	1.99

^1^ Somatic cell score = log_2_ (somatic cell count/1000). ^2^ Milk income = NZD 19 per kg of milk solids, where milk solids are the sum of fat, protein, and lactose yields.

**Table 3 animals-13-01728-t003:** Marginal means (Mean) and standard errors (SE) of daily milk yields and composition in seasonal lactation goats were classified by locomotion score on the test day on three farms based in New Zealand during the 2019–2020 production year.

Test-Day Milk Characteristics	Score 0		Score 1		Score 2		Score 3		Score 4	
	Mean	SE	Mean	SE	Mean	SE	Mean	SE	Mean	SE
Daily yields										
Milk (kg)	3.83 ^a^	0.101	3.84 ^a^	0.101	3.78 ^a^	0.103	3.78 ^a^	0.106	3.56 ^b^	0.121
Fat (g)	120 ^a^	3.33	121 ^a^	3.34	122 ^a^	3.42	120 ^a^	3.54	119 ^a^	4.19
Protein (g)	121 ^a^	3.37	121 ^a^	3.38	121 ^a^	3.43	118 ^a^	3.51	111 ^b^	3.96
Lactose (g)	170 ^a^	7.10	173 ^a^	7.11	173 ^a^	7.16	170 ^a^	7.25	158 ^b^	7.77
Concentrations (%)										
Fat	3.15 ^a^	0.072	3.16 ^a^	0.072	3.18 ^ab^	0.073	3.24 ^b^	0.075	3.40 ^c^	0.082
Protein	3.18 ^a^	0.042	3.17 ^a^	0.042	3.17 ^a^	0.042	3.17 ^a^	0.043	3.16 ^a^	0.047
Lactose	4.52 ^a^	0.035	4.52 ^a^	0.035	4.54 ^ab^	0.035	4.56 ^b^	0.036	4.58 ^b^	0.038
Fat/Protein ratio	0.99 ^a^	0.023	1.00 ^a^	0.023	1.00 ^ab^	0.023	1.02 ^b^	0.024	1.09 ^c^	0.026
Somatic cell score ^1^	9.54 ^a^	0.132	9.58 ^a^	0.132	9.52 ^a^	0.135	9.56 ^a^	0.139	9.76 ^a^	0.162
Milk income ($) ^2^	7.77 ^ab^	0.307	7.82 ^a^	0.307	7.84 ^a^	0.310	7.70 ^a^	0.314	7.25 ^b^	0.337

^1^ Somatic cell score = average log_2_ (somatic cell count/1000). ^2^ Milk income = NZD 19 per kg of milk solids, where milk solids are the sum of fat, protein, and lactose yields. ^a,b,c^ Means with different superscripts within the same row are significantly different (*p* < 0.05).

**Table 4 animals-13-01728-t004:** Marginal means (Mean) and standard errors (SE) of daily milk yields and composition in extended lactation goats were classified by locomotion scores on the test day on three farms based in New Zealand during the 2019–2020 production year.

Test-Day Milk Characteristics	Scores 0		Scores 1		Scores 2		Scores 3		Scores 4	
	Mean	SE	Mean	SE	Mean	SE	Mean	SE	Mean	SE
Daily yields										
Milk (kg)	3.23 ^a^	0.092	3.19 ^a^	0.091	3.20 ^a^	0.092	3.08 ^b^	0.093	2.95 ^c^	0.102
Fat (g)	104 ^ab^	2.89	103 ^ab^	2.84	105 ^a^	2.88	101 ^ab^	2.94	99.0 ^b^	3.30
Protein (g)	105 ^a^	2.92	103 ^ab^	2.88	104 ^ab^	2.91	101 ^ab^	2.96	96.1 ^c^	3.24
Lactose (g)	143 ^a^	5.06	141 ^a^	5.02	141 ^a^	5.05	136 ^b^	5.10	132 ^b^	5.44
Concentrations (%)										
Fat	3.27 ^ab^	0.062	3.26 ^a^	0.061	3.28 ^ab^	0.062	3.32 ^b^	0.062	3.44 ^c^	0.065
Protein	3.27 ^a^	0.028	3.27 ^a^	0.028	3.27 ^ab^	0.029	3.30 ^bc^	0.029	3.32 ^c^	0.030
Lactose	4.46 ^a^	0.019	4.47 ^a^	0.019	4.47 ^a^	0.019	4.48 ^ab^	0.019	4.50 ^b^	0.021
Fat/Protein ratio	1.00 ^a^	0.018	1.00 ^a^	0.018	1.01 ^a^	0.018	1.01 ^a^	0.019	1.04 ^b^	0.019
Somatic cell score ^1^	9.98 ^a^	0.081	10.0 ^a^	0.080	10.0 ^a^	0.081	10.0 ^a^	0.083	10.1 ^a^	0.093
Milk income ($) ^2^	6.63 ^a^	0.221	6.56 ^ab^	0.219	6.57 ^ab^	0.220	6.38 ^bc^	0.223	6.22 ^c^	0.238

^1^ Somatic cell score = average log_2_ (somatic cell count/1000). ^2^ Milk income = NZD 19 per kg of milk solids, where milk solids are the sum of fat, protein, and lactose yields. ^a,b,c^ Means with different superscripts within the same row are significantly different (*p* < 0.05).

## Data Availability

Restrictions apply to the availability of these data. Data were obtained from New Zealand Dairy Goat Co-Operative Ltd. and are available from the authors with the permission of the New Zealand Dairy Goat Co-Operative Ltd.

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
