# Peer review of "The Effect of Lameness on Milk Production of Dairy Goats"

_animals, 2023, doi:10.3390/ani13111728_

Round 1

Reviewer 1 Report

The relation between animal health and production is an important topic for the dairy industry. The results are relevant. However, as the authors describe more analysis will be necessary to be able to produce more accurate data. 

The paper should help to adress the positive effects of claw treatment.The high percentages of clinically lame animals is from an animal welphare am consumers perspective not acceptable.

remarks:

1. In abstract several numbers are mentioned which are not mentioned in the paper. . In line 35 numbers for clinical lame goats are mentioned which I could not find in the paper.

2. In line 38/39 a loss of daily income per goat of 26 and 104 NZD seems not correct. 

3. in the methods it is not clear how the lameness data were correlated with the production data. Are the dates on which the lameness scores are determined. the same on which the milk production data are collected? Please describe more clearly. 

4. Are the differences in milk income corrected for farm level?

5. figure1. It would help if the Y-ax per day would be added.

6. In the discussion it was suggested that the feed intake was lower due to lameness. For accurate financial evaluation it would be desired if the reduction of feed intake would be known.

7. line 331: It should be mentioned that the concentration of fat and protein were elevated. The reduction of milk production was larger. 

Author Response

I have attached my response.

Reviewer 2 Report

Comments

Line 70-72: This part of the sentence is incomplete: “In a more recent British study, the”management of lameness costs farmers between GPB 3.90 and 6.35 (NZD 7.45 and 12.1),…..”

Line 108-110:Mention the composition and ratios of the “total mix ration”. Mention the “supplementary concentrate meal” provided at milking times. Was “fresh cut grass” provided all year round?

Line 111-117: In both the “seasonal” and “extended” lactation groups identify the numbers of animals who had a short duration lameness during her lactation or a long/ repeated episodes of lameness. It would also be of benefit to define a “short duration lameness” and a “long/ repeated episodes of lameness”.

Line 145-146: Please amend the range of days in milk: (ranging from 272 to 3,738 (> 10 years) days in milk at the first test-day event).

Line 151-152: Please amend the range: deviation from the median kidding date in this group ranged from 272 to 3,738 days

Line 211-213: Elaborate on the following finding: The cost of severe lameness was higher 211 in seasonal goats than in extended lactation goats.

Line 228-232: Figure 1: It would be more informative if the animals were divided up into those with an acute, short duration, clinical lameness episode or a chronic lameness

Line 238-239: “Variations were most likely due to management and environmental factors.” Mention these factors in the “Materials and Methods” section. In Line 106-107 in Materials and Methods the author states that “Both groups of goats were housed and managed the same.”

Line 301-302: “if seasonal goats breed each year, the farmer could prioritize the treatment of seasonal goats above extended lactation goats.” Was this investigated by the author on the farms in the study?

Some additional queries that need to be answered:

A.      Mention the frequency of locomotion scoring for an animal during her lactation.

B.      Mention the different types of lameness identified and their incidence rates for both lactation groups.

C.      Mention whether animals where treated if they were identified as being lame. If they were treated, did they remain in the trial?

D.     If animals were treated for lameness, was it for all scores or only certain scores?

E.      How did treating/not treating animals in both lactation groups impact on milk yield?

F.       What criteria were used to include animals in the trial? Were animals who were sick due to another cause included? If included, were these animals treated? 

Author Response

I have attached my response.

Reviewer 3 Report

This study provides a comprehensive analysis of the effect of various degrees of lameness on dairy goat milk yield, composition and profitability. The high portion of goats with extended lactations and, therefore, the high days in milk when lameness scoring was done may not be typical of the general population.

Line   Comment

55 change “due” to “necessary because of”

66 Suggest “hoof rot” for “footrot”

107 State the SAS procedure used eg GLM

126-129 This is reporting a result

189  /1000 is missing

205 change to “the average milk income was 6.10% lower”. 

Author Response

I have attached my response.
